# Investigation of growth curves with different nonlinear models and MARS algorithm in broiler chickens

**Turgay Şengül[1], Şenol Çelik[1]\*, Ahmet Yusuf Şengül[1], Hakan İnci[1], Ömer Şengül[2]**

**1** Dept. of Animal Sci., Faculty of Agriculture, Bingöl University, Bingöl, Türkiye, **2** Dept. of Animal Sci., Faculty of Agriculture, Bursa Uludağ University, Bursa, Türkiye

\* senolcelik@bingol.edu.tr

**Citation:** Şengül T, Çelik Ş, Şengül AY, İnci H, Şengül Ö (2024) Investigation of growth curves with different nonlinear models and MARS algorithm in broiler chickens. PLoS ONE 19(11): e0307037. https://doi.org/10.1371/journal.pone.0307037

**Data Availability Statement:** All relevant data are within the paper and its Supporting Information files.

## Abstract

This study was conducted to determine the live weight model of the broiler chicks by using the most appropriate mathematical growth curves. Live weights were used in broiler chicks grown for 0–6 weeks. Logistics, Gompertz, Weibull, Hossfeld and Von Bertalanffy models and multivariate adaptive regression splines (MARS) data mining algorithm were used to define the live weights of the chickens. In the comparison of the models, the determination coefficient ($R^2$), mean square error (MSE), Akaike's Information Criterion (AIC) and Schwarz Bayesian Information Criterion (BIC) values were used. As a result of the study, it is seen that Gompertz model is the best model to define live weight of the broilers in the Gompertz model, $R^2$, MSE, RMSE, AIC, BIC and growth rates for male broiler were 0.9998, 470.570, 21.681, 68.750, 68.934 and 0.241, respectively. The actual measured live weight values and the weight values estimated by Logistics, Gompertz, Weibull, Hossfeld, Von Bertalanffy models and MARS algorithm are close and in harmony with each other in the graph. However, the weight values estimated from the MARS algorithm are much closer to the observed live weight values. The represent study also demonstrated a very high predictive performance of the MARS data mining algorithm for describing the growth of chicken. In conclusion, MARS algorithm can be a good alternative for breeders aiming at describing the weight-age relationship of broiler chickens.

## Introduction

Growth is a change in the weight and body size of a living organism that occurs during a certain period of time and is a characteristic of great economic importance in animal breeding [1]. The age-related change in the growth of living organism is called the growth curve [2–5]. In the course of reaching the adult weight, which grows to the end of the animal, biological parameters that have an important role in explaining the physiological growth with the growth curve models can be estimated [6].

Growth graph measured as body measurements has been described by mathematical models fitted to growth curves, particularly in poultry [7, 8]. Animal growth is a complex

**Funding:** The author(s) received no specific funding for this work.

**Competing interests:** The authors have declared that no competing interests exist.

physiological and morphological process from hatching to maturity. It is defined as the increase in body weight and organ size per unit of time or age [9]. Growth is an important economic feature in broiler industry and it can be described as an increase in body size per time unit [10]. Body weight of local chickens is an important economic attribute for farmers. This characteristic is hereditary, and therefore, it could be evolved through breeding programs [11].

The application of mathematical model on growth curve will provide a set of parameters that could be used to describe growth pattern overtime. Moreover, it will enable the breeders to expect the weight of animals at a specific age and to detect the phase that associated with the reduction in growth rate [12]. The live weight gain, which occurs in the chicks fed in a certain period, is explained by using the growth curve models. Many studies have been carried out on the determination of the growth curve which best describes growth in broiler chickens and different mathematical models have been tried in these studies [13–18].

Using the weekly individual live weights of 118 broilers from hatch to 49 days old, the growth of the animals was examined in a research. The Rolling-Grey Model (1,1) prediction approach and nonlinear functions (Von Bertalanffy, Gompertz, and Logistic) were employed in the analysis. Analysis revealed that, from the 14th to the 21st day of the experiment, the male chicks had a larger body weight than the females, and that the growth profiles of the two groups of chicks were not parallel [19].

The experimental findings in another study showed that the differential recurrent neural network (DRNN) model relatively well reflected the underlying dynamics of the broiler growth process. Real-time feed intake specifications were made possible by the DRNN-based nonlinear model predictive control (NMPC), which allowed the broiler weights to precisely follow the intended growth curves, which ranged from +12% of the standard curve for broiler chickens to +12%. For the period from day 12 to day 51, the mean relative error between the desired and attained broiler weight was 1.8% overall [20].

In a 42-day timeframe, 640 broiler chicks were sexed in the [21] study. Artificial neural networks (ANNs) were used to assess the predictive accuracy of body weight in place of nonlinear regression models (Logistic, Gompertz, Von Bertalanffy, and Brody). The nonlinear growth models outperformed ANN in fitting the age-weight data, as indicated by the goodness of fit criteria and error measurement statistics. The Ross 308 broiler chicks' body weight fluctuations over time might be predicted by using the Gompertz model.

The growth rate of broiler chickens during the fattening period shows significant differences by weeks. In the first weeks there is a rapid development, a slow growth from a certain point of bending to the end of the fattening period. For this reason, the use of nonlinear mathematical models is the subject. In this study, 5 different growth models (Logistics, Gompertz, Weibull, Hossfeld and Von Bertalanffy models) and MARS algorithm were studied and tried to be evaluated in terms of broiler chickens.

## Materials and methods

### Ethics statement

**Institutional review board statement.** This is an individual study, hence no institutional review or ethics committee is necessary. The study did not include any persons.

### Experimental design

This study was conducted for 6 weeks in a poultry farm of Bingöl University, Faculty of Agriculture, Department of Animal Science. 192 day-old male broiler chicks (Ross PM3) were used as animal material in the experiment. Day-old chicks were obtained from a commercial company. Chicks were housed in multi-storey broiler cages for the first two weeks. Later, birds

were transferred to the litter system. The birds were provided with a space of 12 animals per m$^2$ in the litter system. Wood shavings were used as litter, and trial chambers were set up with wire mesh on the litter. For the first 3 days, 24-hour lighting was applied daily. In later periods, the lighting program was continued as 23L: 1D daily. The chicks were provided with a temperature of around 33–35°C on the first few days, and towards the end of the experiment, this temperature was reduced to 20°C. Broiler chickens were fed as *ad libitum* with feeds containing 24% crude protein and 3000 kcal/kg ME in the first 14-day period, 22% crude protein and 3200 kcal/kg ME from the 15th day to the 42nd day. Drinking water was provided *ad libitum*. Birds were weighed weekly from start to sixth week.

## Statistical analysis

In the study, Logistics, Gompertz, Weibull, Hossfeld and Von Bertalanffy models were used to estimate live weights of broiler chickens. The models used in this study were calculated as following;

The Logistics growth model,

$$Y_t = \frac{A}{(1 + b * \exp(-k * t))} \tag{1}$$

The logistic growth curve used for somatic growth was revealed by Verhulst in 1838 [22]. The Logistics function has a point of inflection at [23],

$$\left( IPT = t_{inf} = \frac{\ln b}{k}, IPW = Y_{tinf} = A/2 \right) \tag{2}$$

The Gompertz growth model [24]

$$Y_t = A * \exp(-b * exp(-k * t)) \tag{3}$$

The Gompertz function has a point of inflection at (IPT and IPW) [23, 25].

$$\left( IPT = t_{inf} = \frac{\ln b}{k}, \ IPW = Y_{tinf} = A/e \right) \tag{4}$$

Weibull function [25],

$$y_t = A - b * \exp\left(-kt^\lambda\right) \tag{5}$$

For Weibull function, age (*IPT*) and weight (*IPW*) at point of inflection [25, 26].

$$IPT = [(\lambda - 1)/k\lambda]^{1/\lambda} \tag{6}$$

$$IPW = A - b * \exp\left(-\left(1 - \frac{1}{\lambda}\right)\right) \tag{7}$$

The Hossfeld Growth Model [27] has the from

$$Y_t = A\left(1 + b_1 t^{-b_2}\right)^{-1}, b_1 > 1 \tag{8}$$

Where $b_2$ serves as a growth rate parameter for a fixed $b_1$ [25]. It has a point of inflection at [25],

$$IPT = \left(\frac{1 + b_2}{b_1(b_2 - 1)}\right)^{-1/b_2} \tag{9}$$

$$IPW = A\left(\frac{b_2 - 1}{2b_2}\right) \tag{10}$$

Bertalanffy growth model is like;

$$y_t = A * (1 - b * \exp(k * t))^3 \tag{11}$$

and it has been presented by Von Bertalanffy (1957) [28] as three-parameter. For Bertalanffy function, age (*IPT*) and weight (*IPW*) at point of inflection respectively,

$$\left(IPT = \frac{\ln(3b)}{k}, \ IPW = 8A/27\right) \tag{12}$$

In these models; A, B, k are growth curve parameters and λ: shape parameter.

A: Asymptotic size or weight, b: A measure of the starting size of the living (fixed), k: Growth constant, t, Time (age), Y: Live weight, A, B, k = Growth curve parameters.

Friedman (1991) [29] primarily developed the MARS algorithm that exhibits the non-linear relationships between a set of predictors (independent variables) and dependent variable. One of the main features of the MARS algorithm is that no stiff assumptions about the functional relationships between the predictors are required. The MARS is a non-parametric regression method that takes a basis for a divide and conquers strategy where the training data sets are split into separate piecewise linear segments (splines) of various slopes. The splines are then smoothly connected to each other and the researcher uses basis functions as piecewise curves to effectively find any linear and/or non-linear effects. The connection points between the pieces are available as "knots" in the MARS. The candidate knots were founded at a random location within the range of each predictor. A stepwise procedure is then used by the algorithm to derive basis functions by taking account of all possible knots and interactions between significant predictors. The MARS predictive model formed by the algorithm at each phase customized the knots and their pairs of basis functions for reducing error variance [30]. The MARS algorithm's equation can be written following as:

$$f(x) = \beta_0 + \sum_{m=1}^{M} \beta_m h_{m(x)} \tag{13}$$

Where the summation is over the *M* terms in the model, and $\beta_0$ and $\beta_m$ are parameters of the model (along with the knots *t* for each basic function, which are also estimated from the data) [31].

M represents the number of basic functions in the current model and $h_{m(x)}$ shows the piecewise linear basic functions which are described in following [29].

$$|x - t|_+ = \begin{cases} x - t, \ x > t \\ 0, \ x \leq t \end{cases} \tag{14}$$

$$|t - x|_+ = \begin{cases} t - x, x < t \\ 0, x \geq t \end{cases} \tag{15}$$

Generalized Cross Validation error is a measure of the goodness of fit that takes into account both the residual error and the model complexity. It is expressed as [32].

$$GCV(\lambda) = \frac{\sum_{i=1}^{N} \left(Y_t - \hat{Y}_t\right)^2}{\left[1 - \frac{M(\lambda)}{N}\right]^2} \tag{16}$$

here n indicates the number of training cases and $M(\lambda)$ is a penalty function for the complexity of the model containing $\lambda$ terms.

In order to determine the best model, goodness of fit criteria such as mean square errors (MSE), determination coefficient ($R^2$), Akaike's Information Criterion (AIC) and Schwarz Bayesian Information Criterion (BIC) were used. The mean error squares is obtained by dividing the error squares by the degree of freedom of the sum of squares. For the most appropriate model, the smallest MSE value is preferred [33, 34].

Shortly, the equations are;

$$MSE = \frac{1}{n} \sum_{t=1}^{n} \left(Y_t - \hat{Y}_t\right)^2 \tag{17}$$

Determination coefficient ($R^2$),

$$R^2 = \frac{\sum_{t=1}^{n} \left(\hat{Y}_t - \bar{Y}\right)^2}{\sum_{t=1}^{n} \left(Y_t - \bar{Y}\right)^2} = 1 - \frac{\sum_{t=1}^{n} \left(Y_t - \hat{Y}_t\right)^2}{\sum_{t=1}^{n} \left(Y_t - \bar{Y}\right)^2} \tag{18}$$

Root Mean Square Error (RMSE),

$$RMSE = \sqrt{\frac{1}{n} \sum_{t=1}^{n} \left(Y_t - \hat{Y}_t\right)^2} \tag{19}$$

Akaike's Information Criterion (AIC),

$$AIC = n \ln\left(\frac{SSE}{n}\right) + 2k \tag{20}$$

Schwarz Bayesian Information Criterion (BIC),

$$BIC = n \ln\left(\frac{SSE}{n}\right) + k \ln(n) \tag{21}$$

[35, 36]. Where, n: the number of observations, SSE: Sum of square errors, k: the number of parameters, $Y_t$: observation value, $\hat{Y}_t$: estimate value.

## Results

The live weight values of broiler were compared with the Logistics, Gompertz, Weibull, Hossfeld and Von Bertalanffy models results are given in Table 1. Parameter A values for male broiler chickens were found to be highest in the Weibull (13500.860 g) and lowest in the Logistics (3136.722 g) growth models. The Von Bertalanffy model (0.885 g) for male broiler chickens had the least value of parameter B, while the Weibull (13436.664 g) model showed the highest value. When comparing the latest maturity of animals in the Logistic model to other growth models, the estimates of the maturing rate K varied from 0.736 (Logistic) to 0.241 (Gompertz).

Table 2 shows the goodness of fit statistics for comparing different non-linear growth functions.

As shown in Table 2, the mean square error (MSE) in the Logistics model was 629.327, RMSE was 25.086, the coefficient of determination ($R^2$) was 0.9997, AIC was 70.357, BIC was 70.802, IPT was 4.912, IPW was 1568.361 in the broiler. In the Gompertz model MSE was found as 470.570, RMSE = 21.681, $R^2$ = 0.9998, AIC = 68.750, BIC = 68.934, IPT = 6.579 and IPW = 2521.578. In the Weibull model MSE was found as 1085.258, RMSE = 32.943, $R^2$ = 0.9996, AIC = 74.723, BIC = 75.117, IPT = 10.556 and IPW = 6851.335. In the Hossfeld model MSE was found as 3182.381, RMSE = 56.413, $R^2$ = 0.9960, AIC = 78.934, BIC = 79.227, IPT = 6.074 and IPW = 2176.271. In the Von Bertalanffy model MSE was found as 821.485, RMSE = 28.662, $R^2$ = 0.9996, AIC = 72.317, BIC = 72.829, IPT = 11.354, and IPW = 6130.963.

According to the results obtained, the most suitable model was determined as Gompertz model with the lowest MSE, AIC, BIC statistics and highest $R^2$ value. A = 6854.354 in the Gompertz model, which means the highest live weight that broiler chickens can achieve in their lifetime. K parameter which gives information about the growth rate of broiler chickens. It means how quickly the live weight observed at the age approaches the adult weight. The value of the k parameter in the Gompertz model was discovered to be 0.241. The ratio of the initial live weight to the adult live weight is represented in the study by the b parameter, which is typically approximated in the models examined. 4.882 is the value of the b parameter as determined by the Gompertz model. The age and weight at which the greatest live weight increase was attained are the twist points. The maximum live weight growth in the Gompertz model is represented by IPW = 2521.578 g and IPT = 6.579 for the week with the highest live weight gain.

In the present study, Gompertz model best predicted live weight growth in broiler chickens. The logistic model is the second alternative model that defines growth after the Gompertz model. The mature live weight (A) predicted the highest in Von Bertalanffy and the lowest in the Hossfeld model. The model with the highest estimate of the k parameter expressing the rate of slowing was Logistics and the least predictive was the Weibull model. However, for the Hossfeld model, the rate of growth is not k, but the parameter $b_2$. The parameter b, which

**Table 1. Growth model parameters for live weight in broiler chickens.**

| Parameters | Logistics | Gompertz | Weibull | Hossfeld | Von Bertalanffy |
|---|---|---|---|---|---|
| A | 3136.722 | 6854.354 | 13500.860 | 8872.315 | 20692.727 |
| B | 37.106 | 4.882 | 13436.664 | - | 0.885 |
| K | 0.736 | 0.241 | 0.005 | - | 0.086 |
| Λ | - | - | 1.941 | - | - |
| b1 | - | - | - | 106.170 | - |
| b2 | - | - | - | 1.963 | - |

**Table 2. Goodness of fit statistics for different non-linear growth models in chicken.**

| Statistics | Logistics | Gompertz | Weibull | Hossfeld | Von Bertalanffy |
|---|---|---|---|---|---|
| $R^2$ | 0.9997 | 0.9998 | 0.9996 | 0.9960 | 0.9996 |
| Adj. $R^2$ | 0.9995 | 0.9997 | 0.9993 | 0.9940 | 0.9994 |
| MSE | 629.327 | 470.570 | 1085.258 | 3182.381 | 821.485 |
| RMSE | 25.086 | 21.681 | 32.943 | 56.413 | 28.662 |
| AIC | 70.357 | 68.750 | 74.723 | 78.934 | 72.317 |
| BIC | 70.802 | 68.934 | 75.117 | 79.227 | 72.829 |
| IPT | 4.912 | 6.579 | 10.556 | 6.074 | 11.354 |
| IPW | 1568.361 | 2521.578 | 6851.335 | 2176.271 | 6130.963 |
| Final body weight | 2164.56 | 2176.04 | 2177.50 | 2138.51 | 2175.33 |

IPT: Point of inflection time, IPW: Point of inflection weight.

shows the ratio of live weight gain to mature live weight after the hatching from the egg, is estimated maximum in Weibull and the lowest in Von Bertalanffy. However, when model harmonization criteria are taken into consideration, it has been determined that Weibull, Hossfeld and Von Bertalanffy models can not define growth better than Gompertz and Logistic models. Diagrams of the growth curves of broiler chickens during the 0–6 week feeding period are presented in Fig 1.

When Fig 1 is examined, it is seen that the values observed and predicted in other models are compatible with each other except Hossfeld model. However, the observed and predicted values in the Hossfeld model seem to have diverged.

As a result, it can be said that it is important to monitor the increase in live weight in commercial broiler chick until the end of the fattening period and it will contribute to the decision making of producers and profitability to growers.

Results of goodness of fit statistics displayed that predictive models constructed using the MARS algorithm provided the nearly highest predictive accuracy for all chickens, respectively. The $R^2$ adj. values from fitting MARS algorithm were found higher than that of the Gompertz, the best non-linear function chosen among the candidate growth models. $R^2$, Adj. $R^2$, MSE, RMSE, AIC and BIC values calculated for the MARS algorithm were found as 0.9999, 0.9998, 411.786, 20.293, 24.986 and 25.232, respectively.

The first, third, and fourth weeks are a significant knot (breaking point) that causes broiler chickens' body weight to grow from the fourth week till delivery when the second and third terms and coefficients in all the analyzed MARS models are taken into account. For example, the bodyweight of a male chicken that is 0 weeks old may be estimated using,

$$y = 326.7385 + 130.654 * \max(0, \text{age} - 1) - 93.33294 * \max(0, 3 - \text{age}) \\ + 292.7998 * \max(0, \text{age} - 3) + 151.775 * \max(0, \text{age} - 4)$$

$$y = 326.7385 + 130.654 * \max(0, 0 - 1) - 93.33294 * \max(0, 3 - 0) \\ + 292.7998 * \max(0, 0 - 3) + 151.775 * \max(0, 0 - 4)$$

$$y = 326.7385 + 130.654 * \max(0, -1) - 93.33294 * \max(0, 3) + 292.7998 * \max(0, -3) + 151.775 * \max(0, -4) \\ = 46.74 \text{ g}$$

Similarly, the live weight of a male chicken with age = 5 can be calculated as follows.

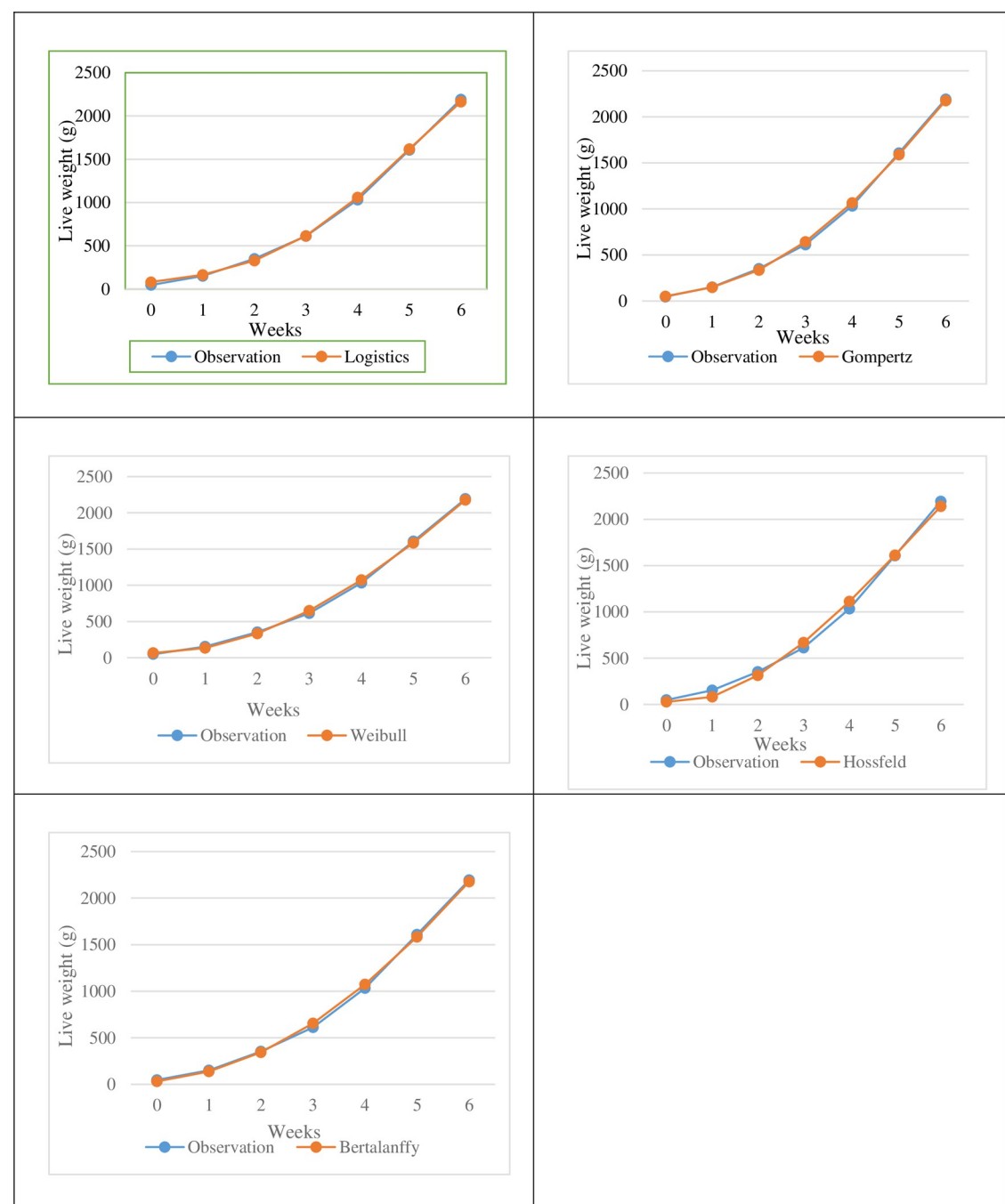

**Fig 1. Estimated growth curves of broiler chickens.**

$$y = 326.7385 + 130.654 * \max(0, 5 - 1) - 93.33294 * \max(0, 3 - 5) + 292.7998 * \max(0, 5 - 3)$$
$$+ 151.775 * \max(0, 5 - 4)$$

$$y = 326.7385 + 130.654 * \max(0, 4) - 93.33294 * \max(0, -2) + 292.7998 * \max(0, 2) + 151.775 * \max(0, 1)$$
$$= 1586.73$$

Here, the effect on body weight is 130,654 g when age>1. When age $\leq$ 3, there is no effect on body weight, when age>3, body weight will increase by 292.7998 g. If age>4, live weight will increase by 151.775 g.

The MARS algorithm's projected body weights are shown in Fig 2. The spline graph illustrates that for male broiler chicks, the MARS algorithm has a cut point accessible at six weeks of age.

The growth curves estimated for broiler chickens with the help of all nonlinear models used are shown in Fig 3.

Fig 3 displays the overall differences in fit between the five models in broiler chicken. Over the various time periods, the Gompertz model's fit of the curves a little differed from that of other models. This is a crucial factor to take into account while selecting the right model. Short interval weight discrepancies between the anticipated and actual weight are more desirable for models than lengthy interval weight variances.

For broiler chickens, the graphs predicted from the MARS algorithm and the Gompertz growth model, which were found appropriate together with the observed values, are presented in Fig 4.

In conclusion, Table 3 presents the observed body weights of male broiler chickens from birth to six weeks together with the anticipated body weights derived from the MARS algorithm and nonlinear growth models employed in this work.

## Discussion

[37] used Logistic, Gompertz, Von Bertalanffy, Morgan-Mercer-Flodin (MMF) and Weibull models to estimate the optimal growth model for broiler chickens and they recommended MMF, Weibull and Gompertz models. In the Gompertz model, the A, b, and k parameters

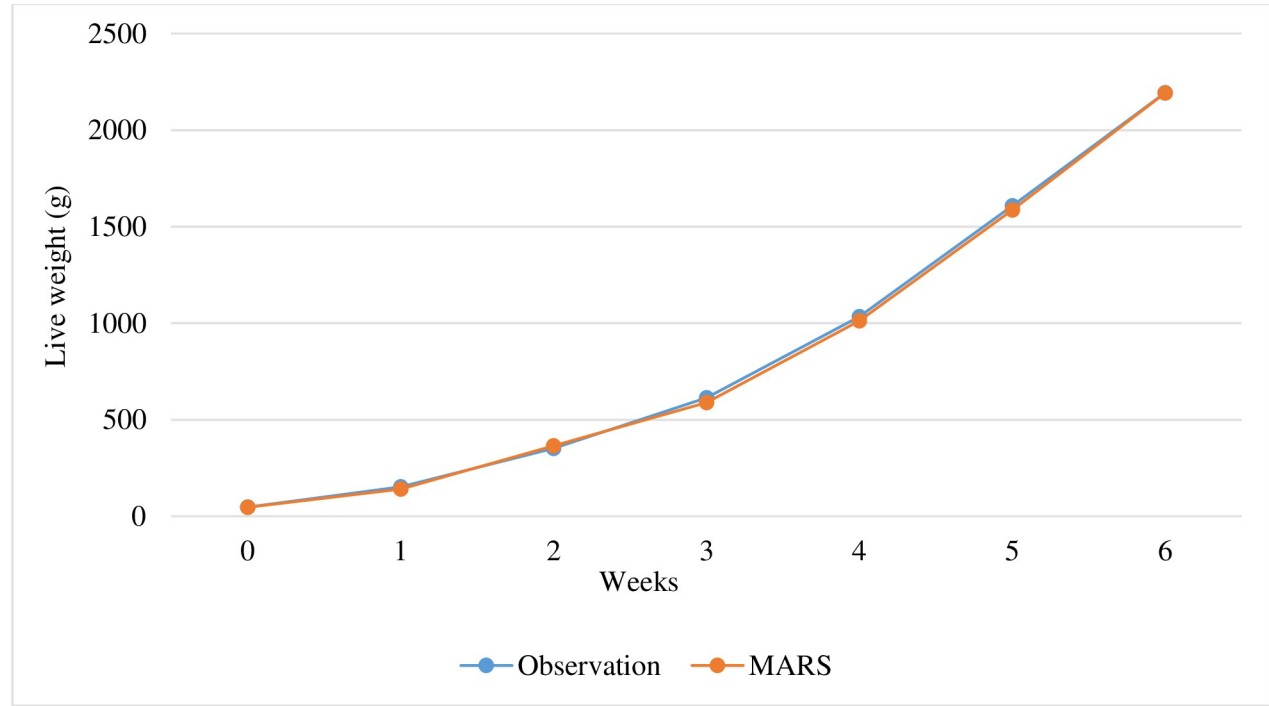

**Fig 2. Predicted body weights (kg) as a function of time (age in weeks) obtained from MARS algorithm.**

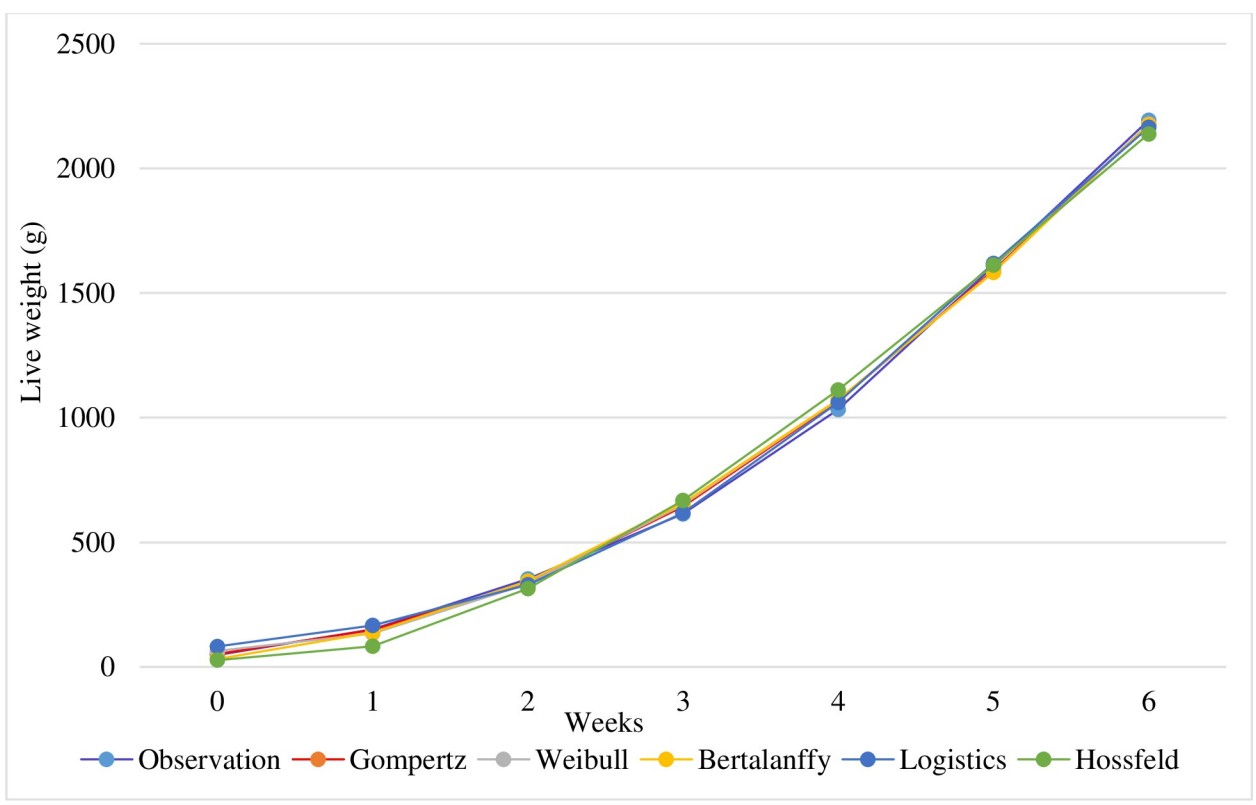

**Fig 3. Estimated all growth curves together of broiler chickens.**

were as follows; 6282.347, 5.313 and 0.268 for males, 5453.802, 4.916 and 0.265 for females. The reported results of b and k parameters were similar to those obtained with this study.

[15] examined the Bertalanffy, Gompertz and Logistic models on the growth of broiler chickens and found that the most suitable model was the Gompertz model. In the Gompertz model, the researcher evaluated the asymptotic weight (A), b and mature growth rate (k) values; 2939, 4.60 and 0.85, respectively. $R^2$, MSE and r values were 0.9997, 408.14 and 0.9997, respectively. Although there were differences between the results reported for the A and k parameter coefficients and the findings of this study, similar results were obtained for the values of the goodness of fit criteria values.

[16] evaluated the growth curves of broiler chickens in different settlement periods with the Gompertz model. In the study, the asymptotic weights (A) of broiler chickens were as follows; 4198.46, 3807.45 and 3999.92 g, and the growth rates (k) were estimated as 0.055, 0.058 and 0.052, respectively. The reported results show that the asymptotic weight is lower and the growth rate is higher when compared with the findings of this study.

[38] investigated Gompetz and Logistic growth models in slow-growing chicken genotypes. According to the Gompertz model, the asymptotic weight of chickens in 4 different genotypes (A) were 3725.34–6496.47; b parameter 5.36–5.84, mature growth rate (k) 0.1288–0.1525; the age of inflection point 11.54–13.99; they found the weight of the inflection point between 1370.58–2390.18. According to the logistic model, the asymptotic weight (A) of the different genotypes were 2133.33–3635.00; b parameter 49.38–62.52, mature growth rate (k) 0.3734–0.3949; the age of inflection point 10.02–11.04; they determined the weight of the inflection

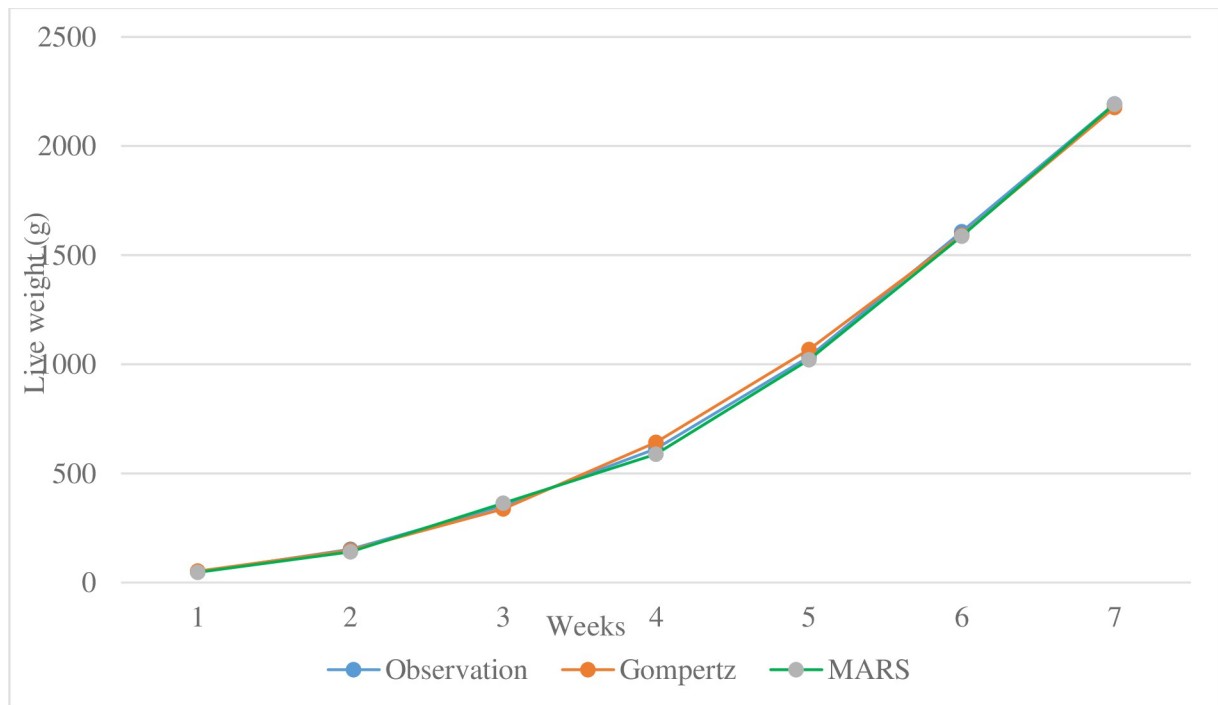

**Fig 4. Predicted body weights (kg) as a function of time (age in weeks) obtained from MARS algorithm and Gompertz model.**

point between 1066.67–1817.50. In this study, the adult growth rate (k) was higher and the b-parameter and inflection point values were lower than those of both models.

In a study by [17], growth models in chickens were compared with Gompertz, Logistics and Von Bertalanffy models. As a result of the most suitable Gompertz model, r, $R^2$ and MSE statistics were 0.9994, 0.9995 and 1068.6, respectively. The parameters A, b and k were estimated to be 2973, 4.67 and 0.05. The findings were different from those of this study, but the same model was estimated.

[18] worked with Gompertz, Logistics and Von Bertalanffy models on growing broiler chickens. Researchers have suggested that the Gompertz model were more appropriate than the other models. According to Gompertz model, A, b and k values of model parameters were 3530 g, 4.723 and 0.0302, respectively, $R^2$ was obtained as 0.940. The reported A and b parameters and $R^2$ values were similar to those of this study.

[39] investigated the growth rate of broiler chickens with Gompertz and Logistic models. They reported that both models showed good results, but Gompertz model was better than

**Table 3. Observed average body weights (g) and estimated body weights by different methods.**

| Weeks | Observed weight | Logistics | Gompertz | Weibull | Hossfeld | Bertalanffy | MARS |
|---|---|---|---|---|---|---|---|
| 0 | 47.68 | 82.32 | 51.95 | 64.20 | 27.19 | 31.79 | 46.73 |
| 1 | 151.76 | 167.02 | 148.01 | 134.94 | 82.79 | 138.11 | 140.07 |
| 2 | 352.37 | 329.50 | 336.88 | 333.90 | 314.36 | 343.78 | 364.06 |
| 3 | 613.21 | 617.18 | 642.79 | 649.68 | 668.09 | 656.31 | 588.05 |
| 4 | 1033.27 | 1061.10 | 1067.8 | 1070.93 | 1111.73 | 1072.64 | 1021.00 |
| 5 | 1607.70 | 1619.11 | 1590.9 | 1584.89 | 1611.92 | 1583.31 | 1586.73 |
| 6 | 2193.10 | 2164.56 | 2176.04 | 2177.50 | 2138.51 | 2175.33 | 2191.96 |

Logistic. According to Gompertz model, A, b and k parameters were 4364.53, 4.62 and 0.356; the $R^2$ and MSE values of statistics were 0.9995 and 729.53, respectively.

[40] tested Brody, Von Bertalanffy, Logistics, Gompertz and Richards models and found that the most suitable model was Gompertz in a study of the broiler genotype. In the Gompertz model, the researchers used $R^2$, RMSE, AIC and BIC statistics in male chicks and the values were 0.9993, 20.76, 79.385 and 76.385, respectively, while the females had 0.9982, 24.12, 83.048 and 80.389. Even though reported AIC and BIC values were different than to the results of this study, $R^2$ and RMSE values were similar. In terms of the most favorable model, the result was similar to this study.

[41] examined different growth models in three different broiler genotypes. According to the proposed Gompertz model, parameters A and b were 6401–6480 and 4.428 and 4.498, respectively. The results of this study showed a difference in the A parameter and a similarity in the b parameter. According to the Gompertz model, $R^2$ and AIC values in three different genotypes in terms of goodness of fit criteria were; 0.9971–0.9983 and 435.36–457.21 in females, 0.9977–0.9984 and 449.68–463.11 in males, respectively. Reported results for $R^2$ values showed similarity, however. AIC values were different than this study.

[42] reported that Gompertz model was the best model to define growth in broiler chickens. The results of this study were partially similar with the findings of the study in which the maximum live weight was reported as A = 3623, b parameter 31.73 and mature growth rate (k) = 0.053, $R^2$ = 0.9970 and AIC = 648.

In other study, the lowest predicted asymptotic weight in hens and cocks (1652.3 and 2356.9 g, respectively) was obtained using the logistic model, while the highest asymptotic weight was estimated using the Richards model for hens (2012.8 g) and using the Von Bertalanffy model for cocks (3011.3 g) [43].

In a different research, 823 chickens from a paternal pure Arian broiler line had a total of 5584 body weight data. Each group's body weight data was fitted independently to five nonlinear growth curve functions: Von Baretanalffy, Gompertz, Lopez, Richards, and Logistic. Richards's function was shown to be the best for all groups based on the goodness of fit criteria, followed by Logistic, Gompertz, Lopez, and Von Bertalanffy (based on the AIC criterion) [44]. In terms of the most advantageous model, the results differed from this study.

In order to determine the growth characteristics for commercial broiler chickens in Indonesia, a total of 1,570 samples were collected. From 0 to 7 days of age to 1 to 5 weeks of age, the samples were weighed. The measured body weights were fitted using a nonlinear Gompertz growth model. The mature live weight's asymptotic value (A) ranged from 3.733 to 5.044 kg, the growth turning point (B) from 4.499 to 4.561, and the growth rate constant (K) from 0.049 to 0.059 kg/week. The inflection points ranged from 25.292–30.970 days and 1.373–1.855 kg for inflection age and inflection weight, respectively. These results are based on the growth parameters [45]. The outcome was comparable to our investigation in terms of the most advantageous model.

It was modelled the data on body weights at 1, 7, 14, 21, 28, 35, and 42 days of 4 commercial broiler genotypes reared in Ghana using the Gompertz and polynomial growth functions. Gompertz function predicted growth better for broiler chicken than the polynomial. The Gompertz growth functions had a greater goodness of fit than the polynomial functions when the $R^2$ and residual errors for each broiler genotype were analyzed. Conversely, the Gompertz function's residual errors were less than the polynomial function's [46].

According to the results of studies conducted by different researchers, the Gompertz model was generally recommended for growth models of broiler chickens. This result supports the result of this study, and Gompertz model was found as the most suitable model in this study. The logistic model can be considered as an alternative model to the Gompertz model.

## Conclusion

In this study, the growth rate of male broiler chickens from the beginning to the experiment to the age of 6 weeks was compared. The 5 nonlinear growth models and MARS algorithm examined were found suitable for live weight data of chickens. As a result, it was determined that the best nonlinear growth model explaining the weight-age variation of broiler chickens was the Gompertz model. Because the Gompertz model was selected as the best fit model for growth in male chicken chickens compared to other candidate models based on the highest $R^2$ and Adj. $R^2$ values and lowest MSE, RMSE, AIC and BIC values. Furthermore, this study also exhibits the potential of the MARS data mining algorithm for describing the body weight-age relationship in broiler chickens.

## Supporting information

**S1 Table.**
(DOCX)

## Author Contributions

**Conceptualization:** Turgay Şengül.

**Data curation:** Şenol Çelik.

**Formal analysis:** Şenol Çelik.

**Investigation:** Şenol Çelik, Ahmet Yusuf Şengül.

**Methodology:** Turgay Şengül, Şenol Çelik.

**Supervision:** Hakan İnci.

**Validation:** Ömer Şengül.

**Visualization:** Şenol Çelik.

**Writing – original draft:** Turgay Şengül, Şenol Çelik.

**Writing – review & editing:** Ahmet Yusuf Şengül, Hakan İnci, Ömer Şengül.

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
