## [Decision Letter · Decision Letter 0]

21 May 2024

PONE-D-24-10742Investigation of growth curves with different nonlinear models and MARS algorithm in broiler chickensPLOS ONE

Dear Dr. Çelik,

Thank you for submitting your manuscript to PLOS ONE. After careful consideration, we feel that it has merit but does not fully meet PLOS ONE’s publication criteria as it currently stands. Therefore, we invite you to submit a revised version of the manuscript that addresses the points raised during the review process.

We look forward to receiving your revised manuscript.

Kind regards,

Hana Maria Dobrovolny, Ph.D

Academic Editor

PLOS ONE

Journal Requirements:

https://doi.org/10.1007/s11250-021-02700-8

In your revision ensure you cite all your sources (including your own works), and quote or rephrase any duplicated text outside the methods section. Further consideration is dependent on these concerns being addressed.

"No authors have competing interests"

5. Please provide a complete Data Availability Statement in the submission form, ensuring you include all necessary access information or a reason for why you are unable to make your data freely accessible. If your research concerns only data provided within your submission, please write "All data are in the manuscript and/or supporting information files" as your Data Availability Statement.

Additional Editor Comments:

Please address the reviewer's comments. In addition, please ensure that units are included every time parameter estimates are presented (in the table and in the text). There should also be estimates of the uncertainty of the parameter values.

Reviewers' comments:

Reviewer's Responses to Questions

**Comments to the Author**

1. Is the manuscript technically sound, and do the data support the conclusions?

Reviewer #1: Yes

Reviewer #2: Partly

2. Has the statistical analysis been performed appropriately and rigorously? 

Reviewer #1: Yes

Reviewer #2: Yes

3. Have the authors made all data underlying the findings in their manuscript fully available?

Reviewer #1: Yes

Reviewer #2: No

4. Is the manuscript presented in an intelligible fashion and written in standard English?

Reviewer #1: Yes

Reviewer #2: Yes

5. Review Comments to the Author

Reviewer #1: 1. Graphical abstract to be add. it gives additional understanding of the process flow of the work

2. Modeling section to be improve with all the parameter expansion in the text

3. Author advised to refer the latest article for citation to get current trends/gab on this field work

4. Provide the justifications how all the proposed approaches given approximately same level of outcomes. If possible use other representations to express the outcomes of the various methodology results.

Reviewer #2: The authors have done a good research work by comparing different growth curve models. The authors are recommended to consider the comments.

1. The ethical clearance details are given to carry out the chicken study.

2. Before starting the experiment, the breeding conditions of the animals were not clearly mentioned.

3. In table 3, the body weight is mentioned as 0.00 for hossfeld model. How can it be possible?

4. In table 2, regression values are mentioned as normal but the body weight data are not clear.

5. Based on the graphical representation, all the models end at the same endpoint. Based on this results, how can you conclude that MARS algorithm and gomertz are good?

6. PLOS authors have the option to publish the peer review history of their article (what does this mean?). If published, this will include your full peer review and any attached files.

Reviewer #1: No

Reviewer #2: No

---

## [Author Response · Author response to Decision Letter 0]

12 Jun 2024

RESPONSE LETTER

This document is a response letter on the manuscript # PONE-D-24-10742 entitled" Investigation of growth curves with different nonlinear models and MARS algorithm in broiler chickens". 

Dear Editor,

Many thanks for sharing valuable comments of you and reviewers with us on improving the manuscript # PONE-D-24-10742 entitled "Investigation of growth curves with different nonlinear models and MARS algorithm in broiler chickens". We are happy that the manuscript will be acceptable for evaluation in "Plos One". We have given answers to all comments of the reviewers evaluating our manuscript using red, green and blue color fonts. Also, red (Reviewer 1), green (Reviewer 2), and blue (Editor) color font on the revised manuscript has been used for indicating all the corrections made by Reviewers.

With Best Regards

Assoc. Prof. Şenol Çelik 

Corresponding author

COMMENTS FOR THE AUTHOR

Reviewer 1

Recommend

1. Graphical abstract to be add. it gives additional understanding of the process flow of the work

Answer 

"The actual measured live weight values and the weight values estimated by Logistics, Gompertz, Weibull, Hossfeld, Von Bertalanffy models and MARS algorithm are close and in harmony with each other in the graph. However, the weight values estimated from the MARS algorithm are much closer to the observed live weight values", the statement added.

Recommend

2. Modeling section to be improve with all the parameter expansion in the text

Answer 

Modeling section was improved with all the parameter expansion in the text.

Recommend

3. Author advised to refer the latest article for citation to get current trends/gab on this field work 

Answer 

Both in the introduction and in the discussion section, current studies were included.

Done (Line 55-74 and Line 355-377).

Recommend

4. Provide the justifications how all the proposed approaches given approximately same level of outcomes. If possible use other representations to express the outcomes of the various methodology results.

Answer

Data checked and re-analysed. Some changes that occurred as a result of the analysis are marked in green. The results for all methodologies used were close to each other. However, goodness-of-fit tests showed that the Gompertz model gave better results than the others among the growth models. When the MARS algorithm was included, the MARS algorithm gave the best result when all models were compared. In addition, the graph of the growth models together is shown in Figure 3. In addition, the graph of the MARS algorithm and Gomperz model together, which gave the best prediction results with the observed values, is shown in Figure 4.

Reviewer 2

The authors have done a good research work by comparing different growth curve models. The authors are recommended to consider the comments.

Recommend

1. The ethical clearance details are given to carry out the chicken study.

Answer

Done. 

It is given in the Materials and Methods section. "Institutional review board statement: This is an individual study, hence no institutional review or ethics committee is necessary. The study did not include any persons".

Recommend

2. Before starting the experiment, the breeding conditions of the animals were not clearly mentioned.

Answer

Necessary information about the breeding and reproduction conditions of animals was written (Line 87-100).

Recommend

Line 13-14: It would have been good to highlight the recommended solutions to current problems within "Horticultural Crop Secondary Metabolism".

Answer

Highlighted in the proposed solutions to existing problems within the scope of "Secondary Metabolism in Horticultural Plants" (Line 13-14).

Recommend

3. In table 3, the body weight is mentioned as 0.00 for hossfeld model. How can it be possible?

Answer

The analysis was redone and the predicted first week body weight for the Hossfeld model is 27.19, not 0.00. The first week body weight was corrected as 27.19.

Recommend

4. In table 2, regression values are mentioned as normal but the body weight data are not clear.

Answer

Since the results changed after the data were re-analysed, the information in Table 2 has been rearranged. Final body weight information was added next to the regression values.

Recommend

5. Based on the graphical representation, all the models end at the same endpoint. Based on this results, how can you conclude that MARS algorithm and gompertz are good?

Answer

The results of goodness of fit tests were more decisive. As a result of the goodness-of-fit tests, the MARS algorithm and the Gompertz growth model gave better results. However, in order to make this better understandable in the graph, we have removed the graph of Logistics, Gompertz, Weibull, Hossfeld and Von Bertalanffy models together with the observed values from Figure 2 and added them separately in Figure 3. In addition, we have given the graph of the observed values together with the predicted values from the MARS algorithm and Gompertz growth model in Figure 4.

In addition, as stated by Mr Editor, some minor text overlapping with a previously published study, which can be accessed at ‘https://doi.org/10.1007/s11250-021-02700-8’, has been revised.

---

## [Decision Letter · Decision Letter 1]

28 Jun 2024

Investigation of growth curves with different nonlinear models and MARS algorithm in broiler chickens

PONE-D-24-10742R1

Dear Dr. Çelik,

We’re pleased to inform you that your manuscript has been judged scientifically suitable for publication and will be formally accepted for publication once it meets all outstanding technical requirements.

Kind regards,

Hana Maria Dobrovolny, Ph.D

Academic Editor

PLOS ONE

Additional Editor Comments (optional):

Reviewers' comments:

Reviewer's Responses to Questions

**Comments to the Author**

1. If the authors have adequately addressed your comments raised in a previous round of review and you feel that this manuscript is now acceptable for publication, you may indicate that here to bypass the “Comments to the Author” section, enter your conflict of interest statement in the “Confidential to Editor” section, and submit your "Accept" recommendation.

Reviewer #1: All comments have been addressed

2. Is the manuscript technically sound, and do the data support the conclusions?

Reviewer #1: Partly

3. Has the statistical analysis been performed appropriately and rigorously? 

Reviewer #1: Yes

4. Have the authors made all data underlying the findings in their manuscript fully available?

Reviewer #1: Yes

5. Is the manuscript presented in an intelligible fashion and written in standard English?

Reviewer #1: Yes

6. Review Comments to the Author

Reviewer #1: responce given by author as The results of goodness of fit tests were more decisive. As a result of the goodness-offit

tests, the MARS algorithm and the Gompertz growth model gave better results.

However, in order to make this better understandable in the graph, we have removed

the graph of Logistics, Gompertz, Weibull, Hossfeld and Von Bertalanffy models

together with the observed values from Figure 2 and added them separately in Figure

3 is accepted

7. PLOS authors have the option to publish the peer review history of their article (what does this mean?). If published, this will include your full peer review and any attached files.

Reviewer #1: No

---

## [Editor Report · Acceptance letter]

15 Oct 2024

PONE-D-24-10742R1 

PLOS ONE

Dear Dr. Çelik, 

I'm pleased to inform you that your manuscript has been deemed suitable for publication in PLOS ONE. Congratulations! Your manuscript is now being handed over to our production team.

Kind regards, 

on behalf of

Dr. Hana Maria Dobrovolny 

Academic Editor

PLOS ONE